# Pharmacological Effects of *Agastache rugosa* against Gastritis Using a Network Pharmacology Approach

**DOI:** 10.3390/biom10091298

**Published:** 2020-09-09

**Authors:** Hyeon-Hwa Nam, Joong Sun Kim, Jun Lee, Young Hye Seo, Hyo Seon Kim, Seung Mok Ryu, Goya Choi, Byeong Cheol Moon, A Yeong Lee

**Affiliations:** Herbal Medicine Resources Research Center, Korea Institute of Oriental Medicine, 111 Geonjae-ro, Jeollanam-do 58245, Korea; hhnam@kiom.re.kr (H.-H.N.); centraline@kiom.re.kr (J.S.K.); junlee@kiom.re.kr (J.L.); wnsl1118@kiom.re.kr (Y.H.S.); hs0320@kiom.re.kr (H.S.K.); smryu@kiom.re.kr (S.M.R.); serparas@kiom.re.kr (G.C.); bcmoon@kiom.re.kr (B.C.M.)

**Keywords:** *Agastache rugosa*, network pharmacology, gastro-protective effects, anti-inflammation, target gene network, bioactive ingredients, signaling pathway

## Abstract

*Agastache rugosa* is used as a Korean traditional medicine to treat gastric diseases. However, the active ingredients and pharmacological targets of *A. rugosa* are unknown. In this study, we aimed to reveal the pharmacological effects of *A. rugosa* on gastritis by combining a mice model and a network pharmacology method. The macrophage and gastritis-induced models were used to evaluate the pharmacological effects of *A. rugosa*. The results show that *A. rugosa* relieved mucosal damage induced by HCl/EtOH in vivo. Network analysis identified 99 components in *A. rugosa*; six components were selected through systematic screening, and five components were linked to 45 gastritis-related genes. The main components were acacetin and luteolin, and the identified core genes were AKT serine/threonine kinase 1 (AKT1), nuclear factor kappa B inhibitor alpha (NFKBIA), and mitogen-activated protein kinase-3 (MAPK3) etc. in this network. The network of components, target genes, protein–protein interactions, and the Kyoto Encyclopedia of Genes and Genomes (KEGG) pathway was closely connected with chemokines and with phosphoinositide 3-kinase-Akt (PI3K/AKT), tumor-necrosis-factor alpha (TNFα), mitogen-activated protein kinase, nuclear factor kappa B, and Toll-like receptor (TLR) pathways. In conclusion, *A. rugosa* exerts gastro-protective effects through a multi-compound and multi-pathway regulatory network and holds potential for treating inflammatory gastric diseases.

## 1. Introduction

Gastritis, the acute or chronic inflammation response to irritation in the gastric mucosa, remains a serious medical problem for many people globally [1,2]. Acute gastritis is an inflammatory disease that affects many people worldwide; it is mainly caused by the use of nonsteroidal anti-inflammatory drugs (NSAID), smoking, alcohol consumption, and *Helicobacter pylori* infection [2,3,4]. Alcohol consumption is a leading factor in gastritis and causes excessive inflammation and cell necrosis in the gastric mucosa [1]. Many pro-inflammatory molecules that are released during gastritis can be considered potential therapeutic targets to prevent or treat *H. pylori*-induced gastric diseases [5]. The anti-ulcer drugs such as proton pump inhibits (PPIs), antacids, or H2 receptor antagonists are used in the treatment of gastritis, but it causes side effects including osteoporosis, hypergastrinemia, and a decreased absorption of vitamin B12 [2]. Therefore, in order to develop alternative drugs for the treatment of gastritis with less side effects, it is necessary to search for materials based on traditional medicinal plants and mechanism study.

In Korea, Gwakhyangjeonggi-san has been mainly used to treat disorders of the digestive system, including abdominal pain, diarrhea, common cold, and vomiting [6]. This formula comprises 13 herbal medicines, among which *Agastache rugosa* Kuntze is the main herb [7]. *A. rugosa* is used as a food ingredient, spice, and traditional medicine in Korea [8]. *A. rugosa*, also known as Korean mint, belongs to the Labiatae family and grows in Northeast Asian countries, including Korea, Japan, and China [9,10]. Flavonoids, phenylpropanoids, lignans, and terpenoids have been identified in the aerial parts of this plant, and tilianin, acacetin, and rosmarinic acid were revealed as the main compounds [9,11]. This herb is listed in The Korean Herbal Pharmacopoeia (KHP). It is classified as an aromatic and damp dissolving herb for which the taste is acrid and thermal properties are slightly warm with the channel affiliations entering the spleen, stomach, and the lungs according to Korean traditional medicine theory [11]. In addition, *A. rugosa* can dispel dampness, relieve nausea and vomiting, and cure fungal infections [11,12]. It exerts pharmacological activities such as anti-inflammatory [9], antioxidant [13], anti-photoaging [10,14], antimicrobial, antitumor [15], and melanogenesis and tyrosinase inhibitory [8] properties. In recent years, many studies have reported various effects of *A. rugosa* except for protective effects on gastric disease.

Generally, it is difficult to elucidate the pharmacological effects of traditional herbal medicines due to the synergies among various biomolecules, limitations of experimental applications, and specific mechanism of action [16,17]. Network pharmacology, first proposed by Hopkins in 2008, emphasizes on “multi-compounds, multi-target, and multi-disease” rather than “a-drug, a-gene, a-disease” drug action pattern; it appropriately adheres to the holistic approach of herbal medicines [18,19]. The multi-component–multi-target action mechanisms of many herbal medicines have been demonstrated in network pharmacology-based studies [20,21,22]. Although *A. rugosa* has been used to treat gastric diseases in Korea [12], there is insufficient scientific evidence to determine the active ingredients and molecular mechanism of pharmacological effects on gastritis. Therefore, it is pertinent to evaluate the protective effect and network pharmacology of *A. rugosa.*

Although *A. rugosa* has been used as an important herb in formulae for gastrointestinal disease in Korea, its efficacy has not been scientifically proven until now. Moreover, to verify the effect of herb extract, not only the effect of each ingredient but also the synergy among the many ingredients must be considered. In this study, we elucidate the gastro-protective effects of *A. rugosa* with an in vivo and its mechanisms were described by in vitro and a network pharmacology approach. Thus, we confirmed the therapeutic effects of *A. rugosa* on gastric mucosa damage in a HCl/EtOH-induced gastritis model, as gastric acid and EtOH are well-known agents that induce gastric damage [23]. Then, the mechanism of gastro-protective effect for *A. rugosa* was revealed in connection with its anti-inflammatory effect and its pathway was explained as the interaction between active compounds and potential genes based on the network pharmacology method.

## 2. Materials and Methods

The steps in this study were as follows: (1) preparation of extract and chemical profiling according to data mining of literature about improvement of efficacy of *A. rugosa* on gastric mucosa damage; (2) determination of efficacy of *A. rugosa* extracts using in vitro and in vivo experiments; (3) identification of components of *A. rugosa* by searching against public chemical databases (some components were isolated previously in our laboratory, and the components were selected according to the absorption, distribution, metabolism, and excretion (ADME) criteria); (4) identification of the target human genes of components (selected in step 3) by using open databases; (5) linking of the potential genes to the target disease; (6) determination of protein–protein interactions (PPIs); and (7) identification of pathways of the target disease by using public databases. The workflow is illustrated in Figure 1.

### 2.1. 70% Ethanol Extract of A. rugosa and Chemical Profiling

*A. rugosa* was purchased from Kwangmyongdang Pharmaceutical Co. (Ulsan, Korea). A voucher specimen (no. 2-19-0365) was deposited in the Korean Herbarium of Standard Resources in the Korea Institute of Oriental Medicine (KIOM). *A. rugosa* (1.0 kg) was ultrasonicated in 70% EtOH (*v/v*) for 2 h. The extract was filtered through chromatography paper (46 × 57 cm) and evaporated in vacuo. The yield of the 70% EtOH extract of *A. rugosa* was 18.94% (*w/w*); the extract was stored at 4 °C. *A. rugosa* extract (2.5 mg/mL) was dissolved in 50% methanol and filtered through a 0.2-μm syringe filter. ACQUITY UPLC^®^ CSH^TM^ C18 column (2.1 × 100 mm, 1.7 μm, Waters Corporation, Milford, MA, USA) comprising a photodiode array detector (PDA eλ detector, sample manager with flow-through needle (FTN-H), and quaternary solvent manager (Waters Co.) was used. The mobile phase was 0.05% formic acid in distilled water (A) and acetonitrile (B). The UPLC was run in a gradient mode from 85% A→40% A for 9 min. The flow rate was 0.3 mL/min and injected volume was 2 μL at 35 °C (column temperature). The UV wavelength was monitored from 210 to 400 nm and detected at 330 nm.

### 2.2. Cell Culture

The RAW 264.7 macrophage cell line was obtained from the American Type Culture Collection (ATCC, Rockville, MD, USA). Macrophages were cultured in Dulbecco’s modified eagle medium supplemented with 10% fetal bovine serum and 1% penicillin (100 units/mL)/streptomycin (100 µg/mL) at 37 °C in 5% CO_2_ (SANTO, Sakata, Japan). The cells were pretreated with different concentrations (0, 100, and 200 µg/mL) of *A. rugosa* extract with 1 µg/mL lipopolysaccharide (LPS) for 24 h.

### 2.3. Cell Cytotoxicity and NO Production

Cell viability was determined using a CCK-8 Assay Kit, and the absorbance was measured using an enzyme-linked immunosorbent assay (ELISA) plate reader (Multiscan Spectrum, Thermo Scientific, Vantaa, Finland). Cells were cultured in 96-well plates (1 × 10^6^ cells/well) and treated with different concentrations of *A. rugosa* extract (0, 100, and 200 µg/mL), followed by co-treatment with LPS (1 µg/mL) for 24 h at 37 °C in a 5% CO_2_ incubator. After culture for 24 h, the cell plates were centrifuged at 2500 rpm for 5 min; then, 50 µL of the cell culture supernatant was mixed with 50 µL 1% sulfanilamide and 0.1% N-(1-naphthyl)-ethylenediamine dihydrochloride (NED) and incubated at room temperature for 10 min. The absorbance at 540 nm was measured by an ELISA reader. NO production in the samples was determined by sodium nitrite serial dilution standard curve, according to the manufacturer’s protocol.

### 2.4. Level of the Anti-Inflammatory Proteins

To analyze the expression of the pro-inflammatory proteins inducible nitric oxide synthetase (iNOS) and nuclear factor kappa B (NF-κB), cells were grown in 6-well plates (2 × 10^5^ cells/well) and treated with *A. rugosa* extract (0, 100, and 200 µg/mL) and LPS (1 µg/mL) for 2 and 24 h. The cells were lysed using radio-immunoprecipitation assay lysis buffer containing 5 mM ethylenediaminetetraacetic acid (EDTA) (pH 8.0), 50 mM Tris (pH 8.0), 150 mM NaCl, 0.1% sodium dodecyl sulfate, 1% NP-40, and 0.5% sodium deoxycholate for 30 min. The total cell proteins were stored at −80 °C. The expression levels of iNOS and NF-κB p-p65 in the harvested cell proteins were analyzed by using western blotting assay.

### 2.5. Animals

Six-week-old C57BL6 male mice (20 ± 2 g) were purchased from Doo Yeol Biotech (Seocho-gu, Seoul, Korea). All mice were housed according to the animal welfare regulation and approved by the Institutional Animal Care and Use Committee of the KIOM (approval number 20-007). C57BL6 mice were housed in standard mouse cages at a temperature of 23 ± 3 °C, humidity of 50 ± 10%, air ventilation frequency of 10 to 20 times/h, and light intensity of 105 to 300 Lux, with food and water provided randomly.

### 2.6. HCl/EtOH-Induced Gastritis Mouse Model

To evaluate the gastro-protective effects of *A. rugosa* extracts, acute gastritis was induced by HCl/EtOH administration. In total, 18 C57BL6 mice were randomly divided into three groups: (1) normal control mice (*n* = 6), (2) HCl/EtOH control mice (*n* = 6), and (3) mice with HCl/EtOH-induced gastritis treated with 100 mg/kg/day *A. rugosa* (*n* = 6). The HCl/EtOH control group was given normal saline. The *A. rugosa*-treated group was given 100 mg/kg/day extract for five days. The mice were fasted for 24 h prior to the experiments. On day 5, except for the mice in the normal group, all mice received 150 mM HCl with 80% EtOH at 90 min after gastric administration. Gastric tissue samples for determination of ulceration index and histological analysis were collected and immediately fixed with 10% neutral-buffered formalin. Finally, the inner mucosa was photographed using an optical digital camera, and gastric lesions were analyzed using the ImageJ program (https://imagej.net/Downloads, modified on 24 January 2020).

### 2.7. Histological Analysis

Gastric tissue sections (5-µm sections) were stained with hematoxylin and eosin and mounted on glass slides for histological analysis. Periodic acid-Schiff (PAS) staining of gastric mucosa was performed for the detection of mucin. The digital images of PAS-stained mucin-like glycoproteins were obtained using a Leica DM2500 microscope (Leica Microsystems, Germany), with a 200× magnification. The diameter of the portal vein was measured using image measurement software (i Solution DTM).

### 2.8. Searching Chemical Components in A. Rugosa

The chemical components in *A. rugosa* were searched for using public databases in TM-MC: A database of medicinal materials and chemical compounds in Northeast Asian traditional medicine (http://informatics.kiom.re.kr/compound/, updated 3 December 2018) and Korean Traditional Knowledge Portal (http://www.koreantk.com/ktkp2014/, updated 14 December 2019) [24]. Information of chemical components isolated previously from *A. rugosa* in our laboratory was also used for the study [9]. Through the above process, a total of 99 components derived from *A. rugosa* were identified (Appendix A). The information of the searched chemical components, including 2D/3D structures, synonyms, chemical number, and physicochemical properties, were verified using public databases such as PubChem (https://pubchem.ncbi.nlm.nih.gov/ updated 21 February 2020), ChEMBL (https://www.ebi.ac.uk/chembl/, updated 4 April 2019), and ChemSpider (http://www.chemspider.com/, version 2020.0.9.0) [24].

### 2.9. Filtering Strategy for Components

All searched components were screened through an in silico integrative model Absorption, Distribution, Metabolism, Excretion (ADME) using Traditional Chinese Medicine Systems Pharmacology Database [20,24,25,26]. The chemical components were filtered by integrating oral bioavailability (OB) and drug-likeness (DL) criteria [19,27]. OB is an important indicator during the process of drug discovery and development because it indicates the quantity of orally administered drugs that reach the systemic circulation [28]; high OB often plays a key role in drug development [19]. Components with OB ≥ 30% were selected as potential active compounds for further analysis [28]. DL is an ambiguous concept that indicates similarity between components and known drugs. Compounds showing DL property are not drugs but have the potential to become a drug [29]. A DL value above 0.18 indicates that the compound is chemically suitable for drug development [30]. Therefore, according to the Traditional Chinese Medicine Systems Pharmacology Database and Analysis Platform (TCMSP) (TCMSP, http://tcmspw.com/tcmsp.php, version 2.3) criteria, the compounds with OB ≥ 30% and DL ≥ 0.18 were selected as potential active components for further analysis [19].

### 2.10. Target Genes

The target genes linked to the selected chemical compounds were searched for using the STITCH (http://stitch.embl.de/, version 5.0) database. This database provides a platform for exploring known interactions between small molecules and proteins and for protein–protein interactions according to organisms [2,31]. The STITCH confidence score is used to define a set of high-confidence interactions between chemical compounds and proteins [30]. In this study, a final list of genes linked to different chemical components (with a confidence score ≥ 0.7, indicating a high confidence score according to STITCH) were obtained [32]. Gene information, including gene ID and name, was verified in the UniProt database by limiting species to “*Homo sapiens*” [24,30].

### 2.11. Potential Target Genes

Information on gastritis genes was searched for in the open database GeneCards: The Human Gene Database (https://www.genecards.org/, version 4.13) by limiting the species to “*Homo sapiens*” and disease to “gastritis”. This database is a searchable, comprehensive database containing functional genetic information, and predicts human genetic information comprehensively in a user-friendly manner [29]. The aforementioned target genes (Section 2.10) were matched to gastritis-related genes, and the overlapping genes were identified as potential target genes.

### 2.12. Protein–Protein Interaction

A protein–protein interaction (PPI) network can predict protein complexes and functions of unknown proteins. Sophisticated network-based tools have been developed to predict potential disease genes [33,34]. PPI analyses were performed with the STITCH database below 50 for the first interacted genes and the highest confidence score (≥0.900). The default database values were used for PPI analyses. The network edge indicates molecular action, and the active interaction sources selected were text mining data, experimental data, database, co-expression, neighborhood, gene fusion, cooccurrence, and predictions.

### 2.13. Signaling Pathway Analyses

The small molecule–protein and protein–protein networks were visualized using Cytoscape version 3.7.2 (https://cytoscape.org/), and the topology in this network was analyzed in this program. A node denotes a small molecule/gene interaction, an edge denotes an association interaction or any other well-defined relationship, and the degree indicates the number of edge connections. Thus, nodes with a high degree can be key nodes in a network [35]. The functional annotation of genes was analyzed using the Database for Annotation, Visualization, and Integrated Discovery (DAVID) ver. 6.8 (https://david.ncifcrf.gov/home.jsp) and the Kyoto Encyclopedia of Genes and Genomes (KEGG) (https://www.genome.jp/kegg/pathway.html, updated January 14, 2020). Functional annotation analysis of the selected genes from the PPI network was performed using DAVID. KEGG pathways with *p* value < 0.05 were considered statistically significant [29,30].

### 2.14. Statistical Analysis

Ulceration index and inhibition rate were expressed as mean ± SD of three independent experiments. The statistical analysis was followed by Tukey’s multiple comparison tests.

## 3. Results

### 3.1. Chemical Profile of A. rugosa

The chemical profile of 70% EtOH extract of *A. rugosa* at 330 nm is shown in Figure 2. Rosmarinic acid (1), tilianin (2), acacetin-7-*O*-(6″-*O*-malonyl)-β-d-glucopyranoside (3), isoagastachoside (4), acacetin-7-*O*-(2″-*O*-acetyl-6″-*O*-malonyl)-*β*-d-glucopyranoside (5), and acacetin (6) were the main components in 70% EtOH extract of *A. rugosa*, and these components were detected at approximately 4.9, 5.9, 6.7, 7.5, 8.2, and 9.3 min, respectively (Figure 2A). Chemical structures of the main components are represented in Figure 2B. All components except rosmarinic acid were acacetin derivatives classified as flavonoid.

### 3.2. Anti-Inflammatory Effects of A. rugosa on LPS-Activated Macrophages

The cytotoxicity of *A. rugosa* extracts on RAW 264.7 cells were determined using a cytotoxicity assay (Figure 3A). The cells were treated with *A. rugosa* extracts at concentrations of 100 and 200 μg/mL with LPS (1 μg/mL). There was no significant effect on cell viability with *A. rugosa* treatment or LPS activation. The inhibitory effects on inflammatory reactions of *A. rugosa* were measured by NO production in RAW 264.7 cells (Figure 3B). NO production was significantly increased by LPS treatment alone compared to normal controls, but a reduction of NO was observed in the cells pretreated with *A. rugosa* extracts in a dose-dependent manner compared to the control. Expression of inflammatory proteins iNOS and p-NFκB was analyzed by western blot assay. Although the levels of iNOS and p-NFκB p65 were significantly increased in the LPS control, the levels of inflammatory proteins decreased by *A. rugosa* extract treatment in a dose-dependent manner. Thus, the extract was used at concentrations of 100 and 200 μg/mL for subsequent anti-inflammatory activity analysis.

### 3.3. Gastro-Protective Effects of A. rugosa on HCl/EtOH-Induced Gastric Injury

HCl/EtOH administration caused a significant increase in ulceration index in mice and damaged the gastric mucosa, including hemorrhagic erosions compared with the normal control. However, mice treated with *A. rugosa* extract (100 mg/kg/day) for five days showed inhibitory effects on the degree of ulceration and hemorrhagic erosions compared with HCl/EtOH-induced control mice (Figure 4A,E,F). Moreover, in histological analysis, HCl/EtOH administration groups showed changes such as disruption of the surface epithelium when compared with the normal control. Treatment with 100 mg/kg/day of *A. rugosa* ameliorated gastric damage via interruption of epithelium disruption (Figure 4B). In addition, the intensity of PAS staining increased due to increased mucus secretion in the gastric mucosa of mice treated with 100 mg/kg/day of *A. rugosa* (Figure 4C).

### 3.4. ADME Screening of Components

Thirty-two and twenty-three components fulfilled the OB ≥ 30 and DL ≥ 0.18 criteria, respectively. Only four components (diosmetin, luteolin, calycosin, and acacetin) satisfied both OB and DL criteria. Although rosmarinic acid and tilianin did not meet the ADME criteria, they were included as exceptions because they have been reported as the main components as well as biomarkers of *A. rugosa* [11,20,24,36]. Finally, six components of *A. rugosa* were selected as target components; they were classified as five flavonoids (acacetin, calycosin, diosmetin, luteolin, and tilianin) and a phenylpropanoid (rosmarinic acid).

### 3.5. Target Genes Linked to Target Components

The target genes linked to the target components were identified using the STITCH 5.0 database, and the degree of interaction was indicated by a quantified interaction score. A high score indicated a strong interaction between components and genes [37]. A total of 485 genes linked to the components were searched (Appendix A), and 93 target genes with a confidence score ≥ 0.7 were selected. In this process, all genes associated with tilianin with score less than 0.7 were eliminated. Degree of acacetin, calycosin, diosmetin, luteolin, and rosmarinic acid were 10, 11, 5, 72, and 7, respectively. Closeness centralities of five compounds were 0.3244, 0.3266, 0.3139, 0.6783, and 0.3139, respectively. Each betweenness centrality was 0.1214, 0.1593, 0.0222, 0.9622, and 0.1205, respectively. As a node’s degree, betweenness centrality, and closeness centrality becomes larger, the importance of the node in the interaction network increases [38]. Luteolin has the highest degree, closeness centrality, and betweenness centrality in this network. The component–gene network comprised five components and 85 genes, and there were 98 nodes and 105 edges in this network (Figure 5).

### 3.6. Potential Target Genes Selected from Gastritis-Related Genes

A total of 1138 gastritis-related genes were searched in the GeneCard database (Appendix A), and 45 genes overlapped with the aforementioned 93 target genes. These potential genes were linked to five components (Table 1) with 50 nodes and 52 edges, of which five were components and 45 were genes. Acacetin, calycosin, diosmetin, luteolin, and rosmarinic acid were linked to 7, 4, 1, 35, and 4 genes, respectively. Luteolin is the most important node with highest degree, closeness centrality, and betweenness centrality. Six genes, namely cytochrome P450 (CYP1A1), fos proto-oncogene (FOS), jun proto-oncogene (JUN), mitogen-activated protein kinase 1 (MAPK1), mitogen-activated protein kinase 3 (MAPK3), and vascular endothelial growth factor A (VEGFA), were regulated by more than two components in this network.

### 3.7. Protein–Protein Interaction Network

The PPI network of 45 gastritis-related genes with high confidence score ≥ 0.900 and first interactors ≤ 50 genes was obtained using the STITCH DB (Appendix A). In total, 81 genes were searched against 45 target genes (Figure 6). The colors of the lines indicated action types of proteins: yellow green denotes activation, blue denotes binding, sky-blue denotes phenotype, black denotes reaction, red denotes inhibition, navy denotes catalysis, pink denotes posttranslational modification, and yellow denotes transcriptional regulation. The action effects of genes were symbolized as positive, negative, and unspecified.

### 3.8. Signaling Pathway Analyses

We described the pathways of gastritis associated with inflammation. To examine the signaling pathways and functions of core genes, we analyzed gene ontology terms and KEGG pathways. KEGG pathways, including core genes and gastric inflammatory pathways, with a *p*-value < 0.05 were selected. Phosphatidylinositol 3′-kinase-Akt (PI3K-Akt), tumor necrosis factor (TNF), mammalian target of rapamycin (mTOR), Toll-like receptor, chemokine, mitogen-activated protein kinase (MAPK), nuclear factor-kappa B (NF-kB), Hypoxia-inducible factor 1 (HIF-1), and transforming growth factor-beta (TGF-beta) signaling pathways were selected as the main pathways associated with gastritis treatment (Figure 7).

## 4. Discussion

Herbal medicines exert their poly-pharmacological effects by acting on multiple targets using their multi-component framework [39]. To allow for clarity and consistency in pharmacology, there is a need for comprehensive organization and presentation of drug targets [40]. Network pharmacology analysis has been used to confirm the therapeutic effects of medicinal plants in correlations among multi-compounds, multi-proteins/genes, and multi-target synergistic processes [41].

*A. rugosa*, a traditional medicinal plant, has been widely used to treat disorders of the digestive system, including abdominal pain, diarrhea, common cold, and vomiting in Korean population [6]. Previous studies have shown that *A. rugosa* contains multiple components, including flavonoids, phenylpropanoids, and carotenoids and that these phenolic compounds from *A. rugosa* have been reported to have some physiological activities [15]. Studies on the various pharmacological and physiological properties of *A. rugosa* have been reported. Though *A. rugosa* has been frequently used to treat gastric diseases in clinic, its efficacy about gastric disease has not been clearly reported. We investigated the anti-inflammatory effects in macrophages, and the gastric mucosa damage ratio in alcohol induced gastritis mice to identify gastro-protective effects of the *A. rugosa* extract.

Gastric damage caused by EtOH may be associated with the generation of reactive species, reduced cell proliferation, and exacerbated inflammatory response [23]. In this study, we conducted experiments in vitro and in vivo to prove the anti-inflammatory and protective effects of *A. rugosa* on gastric disease induced by HCl/EtOH. Although LPS induced the expression of the pro-inflammatory proteins iNOS and NF-κB in RAW 264.7 cells, pretreatment with *A. rugosa* extracts resulted in a dose-dependent decrease in iNOS and p-NFκB p65 expression (Figure 3). Many scientific literatures have reported on the efficacy of *A. rugosa* extracts; there has been no report on its anti-gastritis effect. We summarized in Table 2 some of the effect of this extract compared to our experiment. Our results indicated that the *A. rugosa* extract exerts anti-inflammatory effects by decreasing NO production in LPS-induced RAW 264.7 cells. In other words, NF-κB, a transcription factor, is activated by LPS in the inflammatory response and can regulate the expression of inflammatory genes such as IL-1β, TNF-α, and iNOS [42].

The gastro-protective effects of *A. rugosa* were determined based on the reversal of mucosal damage, which was measured by ulceration index and analysis of histological changes. According to the ulceration index results, HCl/EtOH induced the appearance of gastric lesions and hemorrhage, and erosions in gastric mucosa, but treatment with *A. rugosa* extracts (100 mg/kg/day) showed more than a 60% gastric mucosal protective effect (Figure 4). The histological results showed that treatment with *A. rugosa* extracts reduced the disruption of epithelial cells in HCl/EtOH-induced acute mucosal damage such as infiltration of inflammatory cells and hemorrhagic lesions. In addition, in mice treated with 100 mg/kg/day of *A. rugosa,* PAS-staining analysis presented intense staining of glycoprotein secretions of gastric wall mucosal glands, which play important roles in protecting newly formed cells [43,44]. The therapeutic effect of *A. rugosa* on gastric ulcer is considered associated with increased anti-inflammatory factor action, gastric mucus secretion, and inhibition of inflammatory cell infiltration. These results show that *A. rugosa* has a protective effect on mucosa injury in gastritis induced by HCl/EtOH.

Rosmarinic acid and five acacetin derivates were detected in 70% ethanolic extract of *A. rugosa* by using HPLC (Figure 2), and 14 flavonoids, including acacetin, diosmetin, and luteolin, were isolated in our laboratory previously [9]. The compounds identified in *A. rugosa* in this study are similar to those identified in previous studies and those deposited in ingredient databases. However, the pharmacological effects of the different components in *A. rugosa* on gastric diseases are unclear. In this study, we screened the main active compounds attributed to the gastro-protective effects by the network pharmacology methodology. We selected five active (target) components (acacetin, calycosin, diosmetin, luteolin, and rosmarinic acid) in *A. rugosa* using ADME screening and interactions of chemicals and genes. The core node component luteolin was linked to 35 gastritis related genes. Acacetin and rosmarinic acid were the major components of *A. rugosa* [11,45] and were associated with eight and four gastritis-related genes, respectively. Acacetin, calycosin, diosmetin, and luteolin were classified as flavonoids, which are known to have protective effects against inflammation [46]. Flavonoid-rich fractions showed greater anti-gastritis activity than anti-ulcer drugs through modulation of the NF-κB signaling pathway in stomach tissues in gastritis-induced mice [47]. Luteolin significantly decreases the size of gastric lesions induced by gastric mucosal exposure to indomethacin [48]. Acacetin and diosmetin inhibited *H. pylori* growth, which causes human gastrointestinal diseases [49]. Kangwan et al. reported that rosmarinic acid acts as a protective agent against gastric ulcers caused by indomethacin [50]. Based on the results of previous studies, five core components were considered to play a critical role in reducing gastritis induced by inflammation. Figure 5 shows the association between inflammation-related genes (e.g., IL1B, IL5, IL13, IL2, IKBKB, FOS, JUN, NO2, etc.) and the five core components.

Further, 45 gastritis-related genes linked to the five components in this network were selected. PPI network analysis was performed with these genes using the STITCH database; the network comprised 81 genes. Pathway mapping of the 81 genes was performed using the DAVID tool, and nine important gastritis-related pathways were selected. The result revealed that the core genes closely related in important pathways were AKT1, IKBKB, IKBKG, IL1B, MAPK3, NFKBIA, RPS6KB1, and TLR4, (Appendix A), which were concerned with chemokine, MAPK, NF-kappa B, PI3K-Akt, TNF, and Toll-like receptor signaling pathways.

Gastritis is preponderantly characterized by inflammation of the epithelial lining of the gastric mucosa in the stomach and is the most common higher gastrointestinal acid-related malady of the digestive tract. Thus, inflammation is a vital process in the response of the body to harmful stimuli and has been considered to play a key role in the development of gastric diseases [28].

In the KEGG network pharmacology analysis, gastritis-related genes were associated with nine pathways. Among them, the PI3K-Akt signaling pathway and NF-kB-mediated signaling pathway were closely correlated with the pathogenesis of gastric disease and intestinal mucosal injury [51]. *A. rugosa* extract effectively reduced NO and NF-κB expression compared with LPS-treated control (Figure 3). In addition, these pathways play important roles in inflammatory response and cell proliferation [2]. The PI3K-Akt signaling pathway plays a critical role in angiogenesis, cell growth, proliferation, metabolism, migration, differentiation, and apoptosis [52]. Figure 7 shows the network of the KEGG pathway and the potential target genes. Twenty-nine potential genes among 45 gastritis-related genes were enriched in PI3K-Akt signaling, and these genes were linked to the five identified components (Table 1). In particular, rosmarinic acid and acacetin inhibited IL2 and IL5, which are involved in PI3K-Akt signaling, thus contributing to the anti-inflammatory effects of *A. rugosa*. PI3K-Akt signaling is activated downstream of tyrosine kinase- and G-protein-coupled receptors in response to growth factors and cytokines (IL-2, -5, and -8). Hyperactivation of the PI3K-Akt signaling pathway during gastric cancer alters the activity of downstream transduction pathways that control overall protein synthesis and translation of individual mRNAs [53]. HIF-1 is specifically expressed in macrophages isolated from human *H. pylori*-positive gastritis biopsies and strongly contribute to the induction of pro-inflammatory genes (IL-6 and IL-1β) and inducible NO synthase in response to *H. pylori* [54]. NF-kB, a family of transcription factors, is the main regulator of inflammatory and apoptosis responses, and it is responsible for dimerization, recognition, binding to DNA, and interactions with inhibitory proteins [55]. Activation of NF-kB results in its translocation to the nucleus, were it upregulates the expression of pro-inflammatory target genes such as NO, COX-2, IL-1, TNF-a, and chemokines and may lead to gastritis [56,57]. Taken together, these studies revealed that effective inhibition of NO expression could reduce inflammation-related pathways and could alleviate the symptoms of gastritis. We demonstrated this phenomenon in our study, where *A. rugosa* extract reduced NO production in vitro and ameliorated gastric damage via interruption of epithelium disruption in vivo (Figure 3C and Figure 4). Previous studies have described the activation of NF-kB complexes in alcohol-induced acute gastric mucosal injury. Alcohol injury in rats led to an increase in gastric ulcer, hemorrhagic erosions, and disruption of epithelial cells and inflammatory genes associated with the NF-kB signaling pathway [43]. An aqueous extract of *Artemisia capillaris* exerted gastrointestinal protective effects on gastric injury by downregulating NF-kB proteins and by reducing cytokine production [58]. Similarly, Kangfuxin, a traditional Chinese medicine formulation, exerted protective effects in an EtOH-induced gastric ulcer model via the PI3K/AKT and NF-kB signal pathways [59].

Based on previous research findings and the association between gastritis-related genes in inflammation system and the putative active components of *A. rugosa,* we explored that the healing mechanisms of *A. rugosa* against gastric mucosa damage was closely related to the inflammatory response in PI3K/AKT and NF-kB signaling pathways. The results of our study would be a useful reference for further researching the mechanisms underlying the effects of *A. rugosa* on gastric diseases. However, it is generally a theoretical science and the experimental verification is still needed. Thus, future studies should elucidate the mechanism underlying the effects of *A. rugosa* extract on gastritis and the signaling pathways regulated by its active components.

## 5. Conclusions

In this study, we confirmed that *A. rugosa* extract exerts gastro-protective effects through amelioration of mucosal damage in a HCl/EtOH-induced mouse model. Moreover, we focused the anti-inflammatory effects on different factors of causing gastritis based on a network pharmacological approach to predict the gastro-protective effects of *A. rugosa*. The results showed that *A. rugosa* is a potential source that can be used in gastritis treatment, and it will contribute to validation of the herbal medicine for treating inflammatory and gastroesophageal diseases. However, it is generally a theoretical science and experimental verification is still needed. Thus, future studies should elucidate the mechanism underlying the effects of *A. rugosa* extract on gastritis and the signaling pathways regulated by its active components. In the future, to verify the pharmacological activity of herbal drugs, we will consider using various public DBs [60,61] consisting of ligand–activity–target relationships.

## Figures and Tables

**Figure 1 biomolecules-10-01298-f001:**
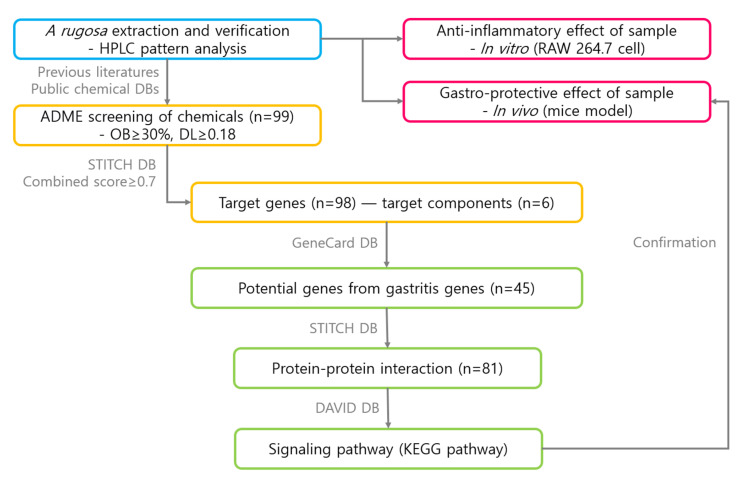
Workflow illustrating the network pharmacological analysis of *A. rugosa*.

**Figure 2 biomolecules-10-01298-f002:**
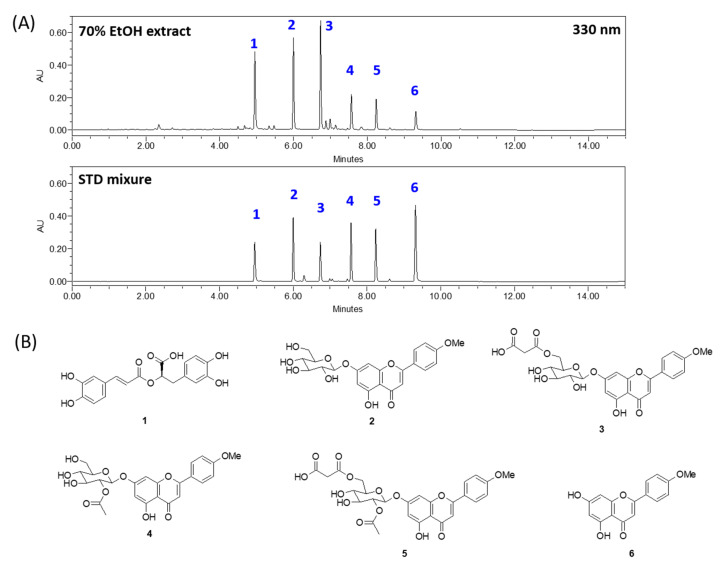
UPLC chromatogram of 70% EtOH *A. rugosa* (**A**) and chemical structures of six main components (**B**).

**Figure 3 biomolecules-10-01298-f003:**
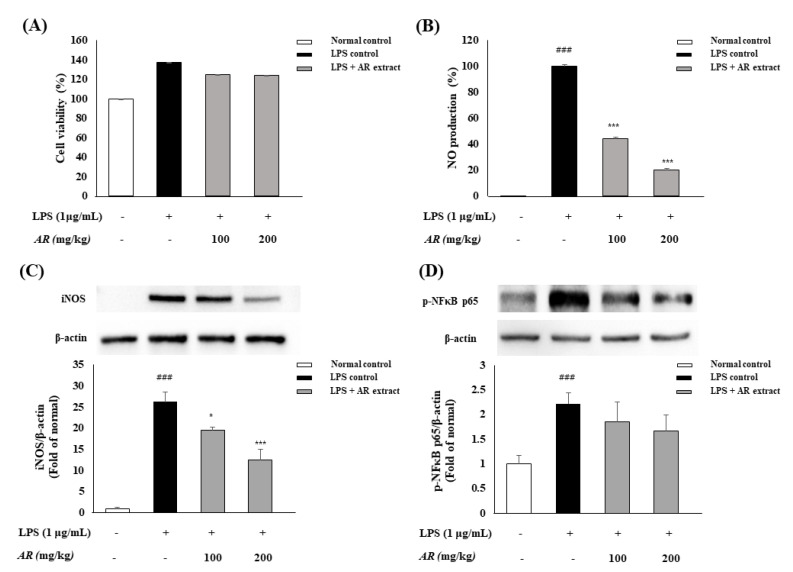
Cell viability (**A**) and NO production (**B**) of LPS (1 µg/mL)-induced RAW 264.7 cells treated with *A. rugosa* extracts (100 and 200 µg/mL) for 24 h: The expressions of iNOS (**C**) and p-NFκB p65 (**D**) proteins for 1 h after the treatment of extracts with LPS in RAW 264.7 cells. Cell viability was determined using a cytotoxicity assay kit, and NO production was measured using the Griess assay. Expression of inflammatory proteins iNOS and p-NFκB p65 was analyzed by western blot assay. Data are means ± standard deviation (SD); ### *p* < 0.001 compared with normal control cells; * *p* < 0.05 and *** *p* < 0.001 compared with LPS control cells.

**Figure 4 biomolecules-10-01298-f004:**
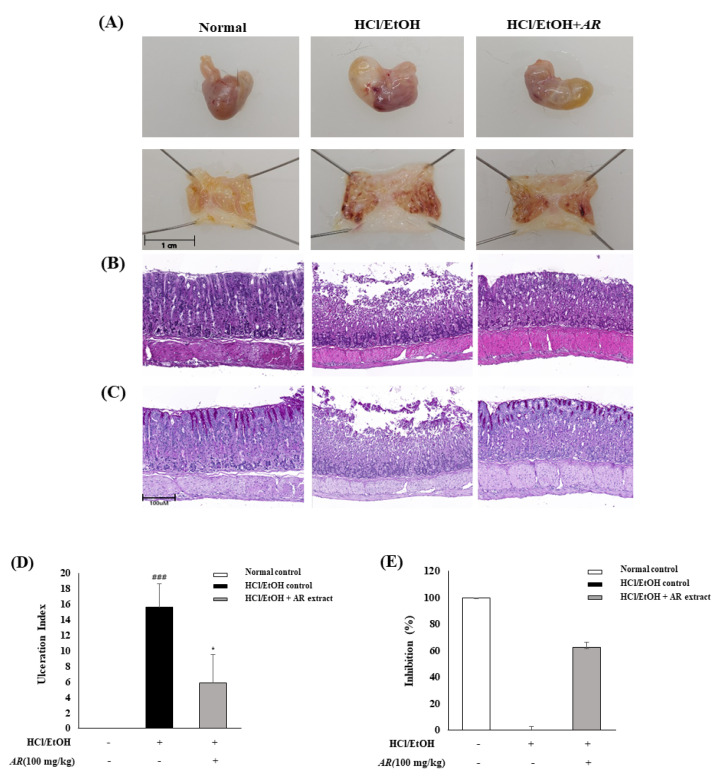
Effects of *A. rugosa* extracts (100 mg/kg/day) on the macroscopic appearance of the gastric mucosa (**A**), ulceration index, inhibition rate (**D**,**E**), histopathological characteristics (**B**), and gastric tissue glycoprotein–Periodic acid-Schiff (PAS) staining (**C**) in HCl/EtOH-induced gastric mucosal lesions in mice. (1) Normal control; (2) HCl/EtOH control; (3) *A. rugosa* extracts 100 mg/kg/day. Results presented as mean ± SEM, *n* = 6. *t*-tests were performed to calculate the statistical significance, ### *p* < 0.001 vs. normal control and * *p* < 0.05 vs. HCl/EtOH control.

**Figure 5 biomolecules-10-01298-f005:**
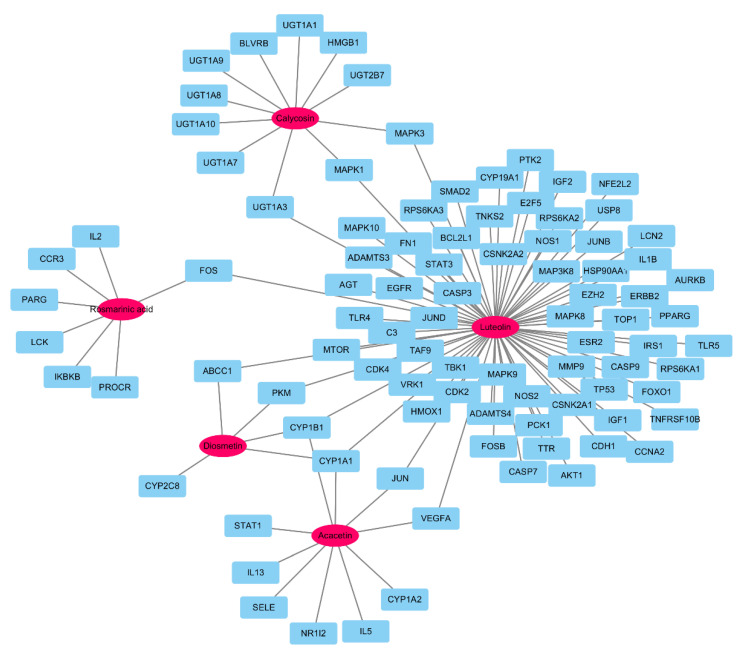
Network of five components (pink oval) and 93 target genes (sky-blue square): this network was comprised of 98 nodes and 105 edges, and luteolin was the most important node because of high score of degree, betweenness centrality, and closeness centrality in this network.

**Figure 6 biomolecules-10-01298-f006:**
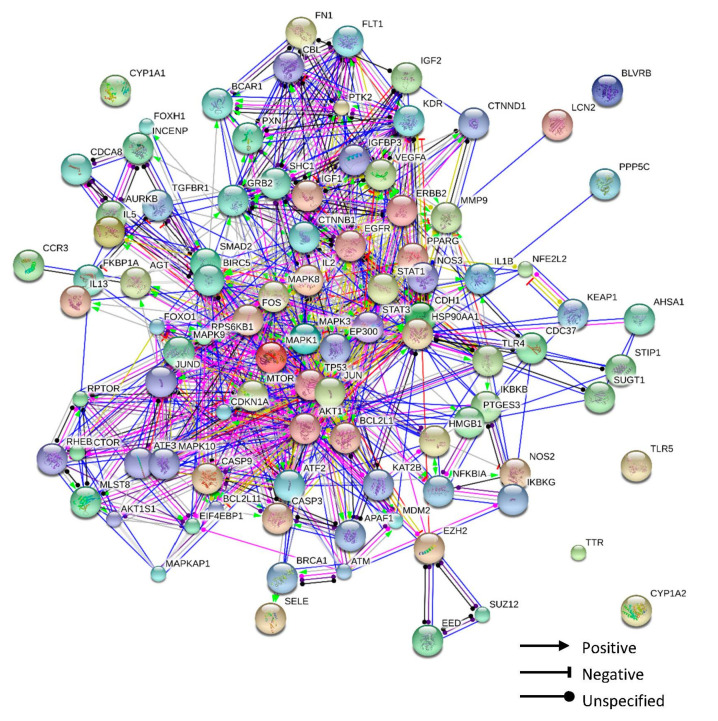
Protein–protein interaction of gastritis-related genes: the shapes of the arrows explain the action effects of interaction of proteins (positive, negative, and unspecified), and the colors of the lines indicate the action types of interaction of proteins.

**Figure 7 biomolecules-10-01298-f007:**
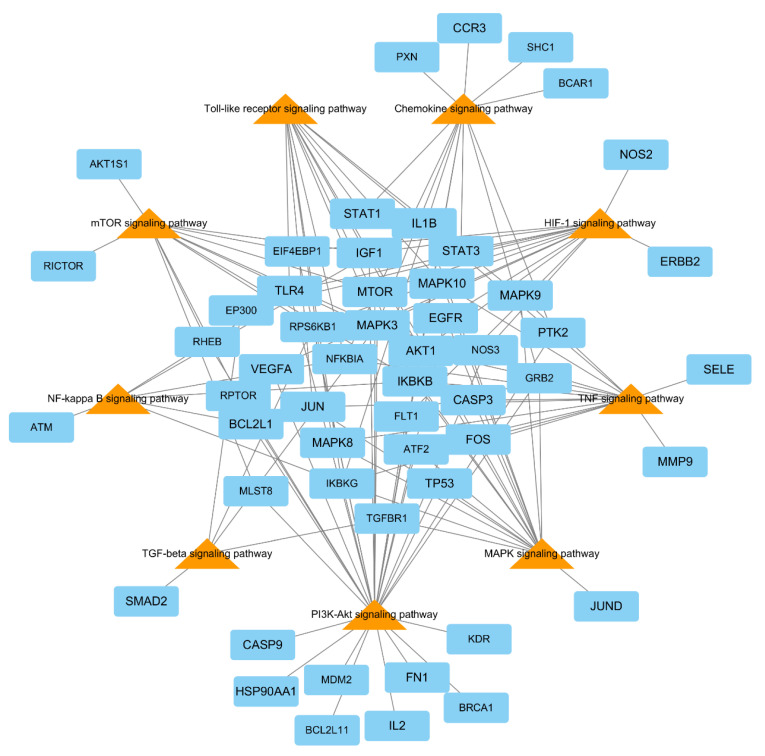
Network of the Kyoto Encyclopedia of Genes and Genomes (KEGG) pathway (orange triangles) and gastritis-related genes (sky-blue squares).

**Table 1 biomolecules-10-01298-t001:** Network of five small molecules and 45 gastritis-related genes.

No.	Components	Genes	Degree	Closeness Centrality	Betweenness Centrality
1	Acacetin	CYP1A1, CYP1A2, IL5, IL13, JUN, SELE, STAT1, VEGFA	8	0.34751773	0.19770408
2	Calycosin	BLVRB, HMGB1, MAPK1, MAPK3	4	0.31210191	0.08120748
3	Diosmetin	CYP1A1	1	0.31612903	0
4	Luteolin	AGT, AKT1, AURKB, BCL2L1, CASP3, CASP9, CDH1, CYP1A1, EGFR, ERBB2, EZH2, FN1, FOS, HSP90AA1, IGF1. IGF2, IL1B, JUN, LCN2, MAPK1, MAPK3, MAPK8, MMP9, MTOR, NFE2L2, NOS2, PPARG, PTK2, SMAD2, STAT3, TLR4, TLR5, TP53, TTR, VEGFA	35	0.67123288	0.94727891
5	Rosmarinic acid	CCR3, FOS, IKBKB, IL2	4	0.31612903	0.11989796

**Table 2 biomolecules-10-01298-t002:** Comparison of the effects of *A. rugosa* extract between previous reports and this experiment.

Effects of *A. rugosa* Extract.	Test Type	Models	Ref.
Tyrosinase and melanogenesis inhibition	In vitro	CCD-986sk, B16F10	[8]
PGE2 inhibition	In vitro	RAW264.7	[9]
Anti-photoaging effect	In vitro	HS68	[10]
Coagulation effect	In vitro	Blood in rabbit	[12]
Antioxidant and antimicrobial effect	In vitro	Six bacterial strains	[13]
Anti-photoaging effect	In vitro	HaCaT keratinocyte	[14]
NO, iNOS inhibition Gastro-protective effect	In vitro In vivo	RAW264.7 Mice	This study

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
