# Peer review of "Pharmacological Effects of Agastache rugosa against Gastritis Using a Network Pharmacology Approach"

_biomolecules, 2020, doi:10.3390/biom10091298_

Round 1

Reviewer 1 Report

This is a resubmitted manuscript. Since significant changes have been made, there could be both follow-up comments and new comments.

(i) Keywords, more representative keywords should be included to summarize the scope of the manuscript. Also, proper selection of keywords could increase the downloading rate and citation of the article.

(ii) Introduction, share literature review (mainly on recent 5 years journal articles) that covers the summary on the methodology, performance, and limitations of existing works.

(iii) The contribution is vague and authors are suggested to summarize the contribution in introduction, preferably in point-form.

(iv) Section 2, Figure 1, please ensure the consistency between the terms in Figure 1 and titles of subsections.

(v) Section 2, proper citations are required instead of presenting the URLs in main text. Share the URLs in list of references.

(vi) Section 2, methodology is not clearly explained. Authors have mentioned there are various databases, each of the databases may have different settings and nature which may require specific consideration and customized approach.

(vii) Figures 3 and 4, legends are missing for the bar charts.

(viii) Under Section 3 Results, the subsections are numbered as 2.x which are typos.

(ix) Discussion, it seems to be in the format of literature review. Please add a table to summarize the results that align with existing works and differ from existing works.

(x) Elaborate the last section Conclusion.

(xi) Elaborate the limitations of proposed work and future research directions.

(xii) Authors may share the visions on the trends of conducting research with public available database. Consider citing the following articles (titles as follows).

- The Concise Guide to PHARMACOLOGY 2019/20: G protein‐coupled receptors

- RIKEN MetaDatabase: A Database Platform for Health Care and Life Sciences as a Microcosm of Linked Open Data Cloud

- The Concise Guide to PHARMACOLOGY 2019/20: Introduction and other protein targets

Author Response

  1. Keywords, more representative keywords should be included to summarize the scope of the manuscript. Also, proper selection of keywords could increase the downloading rate and citation of the article.

→ We thank the reviewer for his/her valuable comment. We have now added three key words “target gene network”, “bioactive ingredient”, and “signal pathway”(page 1 lines 26― 27).

  1. Introduction, share literature review (mainly on recent 5 years journal articles) that covers the summary on the methodology, performance, and limitations of existing works.

→ We thank the reviewer for his/her valuable comment. We have now replaced all the literatures within the latest 5 years and rewritten the introduction partially. However, reference 6 was cited because it could not be altered another. (page 1 line 36, page 2 lines 49―53, 59―64)

  1. The contribution is vague and authors are suggested to summarize the contribution in introduction, preferably in point-form.

→ We thank the reviewer for his/her valuable comment. We have now corrected the author contributions (page 15 lines 472―477)

  1. Section 2, Figure 1, please ensure the consistency between the terms in Figure 1 and titles of subsections..

→ We thank the reviewer for his/her valuable comment. We have now changed Figure 1 to match the subtitles (pages 2, lines 66―68, and 74, Figure 1).

  1. Section 2, proper citations are required instead of presenting the URLs in main text. Share the URLs in list of references.

→ We thank the reviewer for his/her valuable comment. We have now moved the URLs in list of references. (pages 19 and 20, lines 674―687).

  1. Section 2, methodology is not clearly explained. Authors have mentioned there are various databases, each of the databases may have different settings and nature which may require specific consideration and customized approach.

→ We thank the reviewer for his/her valuable comment. We have now described the method parts in detail. (page 3 lines 96―98, 106―107, 110, 115―117, page 4 lines 175―176, page 5 lines 186―187, 189―190, 196―199, 208―209)

  1. Figures 3 and 4, legends are missing for the bar charts.

→ We thank the reviewer for his/her valuable comment. We have now showed the error bars on the Figure 3 and 4. However, some of the small error bars (Figure 3A, 3B, and Figure 4E) were not clearly visible (page 7 Figure 3 and 4)

  1. Under Section 3 Results, the subsections are numbered as 2.x which are typos.

→ We thank the reviewer for his/her valuable comment. We have now changed subsections from 2.X to 3.X. (page 5 line 217, page 6 line 229, page 7 line 250, page 8 line 272, page 9 lines 281 and 301, page 10 line 313, and page 11 line 330)

  1. Discussion, it seems to be in the format of literature review. Please add a table to summarize the results that align with existing works and differ from existing works.

→ We thank the reviewer for his/her valuable comment. We have now rewritten the ‘Discussion’ section, which detailing our results and citing appropriate literatures supported our results. (page 12 line 346―347, 362―363, page 13 lines 364, , lines 367―368, 375, 377, 380―382, 384―388, 401―404, 407―408, page 14 lines 418―420, 422―426, 437―441, and 449―456)

  1. Elaborate the last section Conclusion.

→ We thank the reviewer for his/her valuable comment. We have now rewritten the ‘Conclusion’ section (pages 14 and 15 lines 460―465).

  1. Elaborate the limitations of proposed work and future research directions.

→ We thank the reviewer for his/her valuable comment. We have now clarified the limitation of our research and mentioned the future direction of the study. (page 14 lines 449―456).

  1. (xii) Authors may share the visions on the trends of conducting research with public available database. Consider citing the following articles (titles as follows).

- The Concise Guide to PHARMACOLOGY 2019/20: G proteincoupled receptors

- RIKEN MetaDatabase: A Database Platform for Health Care and Life Sciences as a Microcosm of Linked Open Data Cloud

- The Concise Guide to PHARMACOLOGY 2019/20: Introduction and other protein targets

→ We thank the reviewer for his/her valuable comment. We found these references or databases, unfortunately, the gene for gastritis was found only one gene ‘TLR4’(page 5, line 186 and page 9 line 303). Also, we have now added ‘Discussion’ section (page 12 lines 346―347).

Reviewer 2 Report

The manuscript entitled "Pharmacological effects of Agastache rugosa against gastritis using a network pharmacology approach" reports a study of the pharmacological effects on gastritis using mice model and network pharmacology methods (in silico method, more precisely web tools). I checked some results of the sites following the information of the materials and methods section of the authors. The manuscript is clear, and the computational methods seem technically ok. It is only one point that can be changed. 1 - Conclusions - "our results confirmed the possibility of gastritis treatment with A. rugosa." The results show that A. rugosa is a potential source of potential compounds that can be used as gastritis treatment and it is worth to performer further studies. Please it is suitable to rewrite this sentence.

Author Response

The manuscript entitled "Pharmacological effects of Agastache rugosa against gastritis using a network pharmacology approach" reports a study of the pharmacological effects on gastritis using mice model and network pharmacology methods (in silico method, more precisely web tools). I checked some results of the sites following the information of the materials and methods section of the authors. The manuscript is clear, and the computational methods seem technically ok. It is only one point that can be changed. 1 - Conclusions - "our results confirmed the possibility of gastritis treatment with A. rugosa." The results show that A. rugosa is a potential source of potential compounds that can be used as gastritis treatment and it is worth to performer further studies. Please it is suitable to rewrite this sentence.

→ We thank the reviewer for his/her valuable comment. We have now rewritten conclusion section following this comment (page 14 lines 460―465).

Round 2

Reviewer 1 Report

I have some follow-up comments.

2. Introduction, share literature review (mainly on recent 5 years journal articles) that covers the summary on the methodology, performance, and limitations of existing works.

→ We thank the reviewer for his/her valuable comment. We have now replaced all the literatures within the latest 5 years and rewritten the introduction partially. However, reference 6 was cited because it could not be altered another. (page 1 line 36, page 2 lines 49―53, 59―64)

Follow-up comment: Please summarize the methodology, performance, and limitations of existing works.

3. The contribution is vague and authors are suggested to summarize the contribution in introduction, preferably in point-form.

→ We thank the reviewer for his/her valuable comment. We have now corrected the author contributions (page 15 lines 472―477)

Follow-up comment: This comment has not been addressed. Please summarize the contributions in introduction, preferably in point-form.

5. Section 2, proper citations are required instead of presenting the URLs in main text. Share the URLs in list of references.

→ We thank the reviewer for his/her valuable comment. We have now moved the URLs in list of references. (pages 19 and 20, lines 674―687).

Follow-up comment: Please refer to the journal’s template.

Title of Site. Available online: URL (accessed on Day Month Year).

7. Figures 3 and 4, legends are missing for the bar charts.

→ We thank the reviewer for his/her valuable comment. We have now showed the error bars on the Figure 3 and 4. However, some of the small error bars (Figure 3A, 3B, and Figure 4E) were not clearly visible (page 7 Figure 3 and 4)

Follow-up comment: This comment has not been addressed. Please share legends to indicate the white, black, and grey bars.

9. Discussion, it seems to be in the format of literature review. Please add a table to summarize the results that align with existing works and differ from existing works.

→ We thank the reviewer for his/her valuable comment. We have now rewritten the ‘Discussion’ section, which detailing our results and citing appropriate literatures supported our results. (page 12 line 346―347, 362―363, page 13 lines 364, , lines 367―368, 375, 377, 380―382, 384―388, 401―404, 407―408, page 14 lines 418―420, 422―426, 437―441, and 449―456)

Follow-up comment: Please add a table to enhance the presentation and organization.

11. Elaborate the limitations of proposed work and future research directions.

→ We thank the reviewer for his/her valuable comment. We have now clarified the limitation of our research and mentioned the future direction of the study. (page 14 lines 449―456).

Follow-up comment: It is suggested to move the limitations and future research directions to Section 5. Conclusions.

12. (xii) Authors may share the visions on the trends of conducting research with public available database. Consider citing the following articles (titles as follows).

- The Concise Guide to PHARMACOLOGY 2019/20: G proteincoupled receptors

- RIKEN MetaDatabase: A Database Platform for Health Care and Life Sciences as a Microcosm of Linked Open Data Cloud

- The Concise Guide to PHARMACOLOGY 2019/20: Introduction and other protein targets

→ We thank the reviewer for his/her valuable comment. We found these references or databases, unfortunately, the gene for gastritis was found only one gene ‘TLR4’(page 5, line 186 and page 9 line 303). Also, we have now added ‘Discussion’ section (page 12 lines 346―347).

Follow-up comment: The suggested public available databases are for visions which could be included in conclusions. It does not affect the main theme of authors’ work. Please reconsider the inclusion of the references.

Author Response

  1. Introduction, share literature review (mainly on recent 5 years journal articles) that covers the summary on the methodology, performance, and limitations of existing works.

Follow-up comment: Please summarize the methodology, performance, and limitations of existing works. → We thank the reviewer for his/her valuable comment. We have summarized the methodology, performance, and limitations of this study. (page 1 lines 33―34, 36―40, and page 2 lines 54―55)

  1. The contribution is vague and authors are suggested to summarize the contribution in introduction, preferably in point-form.

Follow-up comment: This comment has not been addressed. Please summarize the contributions in introduction, preferably in point-form.

→ We thank the reviewer for his/her valuable comment. We have now corrected contributions of our experiment (page 2 lines 66―75)

  1. Section 2, proper citations are required instead of presenting the URLs in main text. Share the URLs in list of references.

Follow-up comment: Please refer to the journal’s template.

Title of Site. Available online: URL (accessed on Day Month Year).

→ We thank the reviewer for his/her valuable comment. We have now corrected the URLs according to reviewer’s comment. (page 20, lines 695―709).

  1. Figures 3 and 4, legends are missing for the bar charts.

Follow-up comment: This comment has not been addressed. Please share legends to indicate the white, black, and grey bars.

→ We thank the reviewer for his/her valuable comment. We have now replaced Figure 3 and 4. (pages 7 and 8)

  1. Discussion, it seems to be in the format of literature review. Please add a table to summarize the results that align with existing works and differ from existing works.

Follow-up comment: Please add a table to enhance the presentation and organization.

→ We thank the reviewer for his/her valuable comment. We have now added Table 2. (page 13, lines 373―375, and 381 Table 2)

  1. Elaborate the limitations of proposed work and future research directions.

Follow-up comment: It is suggested to move the limitations and future research directions to Section 5. Conclusions.

→ We thank the reviewer for his/her valuable comment. We have now moved the limitations and future research directions to Section 5. (page 15 lines 476―481).

  1. (xii) Authors may share the visions on the trends of conducting research with public available database. Consider citing the following articles (titles as follows).

- The Concise Guide to PHARMACOLOGY 2019/20: G proteincoupled receptors

- RIKEN MetaDatabase: A Database Platform for Health Care and Life Sciences as a Microcosm of Linked Open Data Cloud

- The Concise Guide to PHARMACOLOGY 2019/20: Introduction and other protein targets

Follow-up comment: The suggested public available databases are for visions which could be included in conclusions. It does not affect the main theme of authors’ work. Please reconsider the inclusion of the references.

→ We thank the reviewer for his/her valuable comment. We have now omitted from this work and instead considered as a vision for the next study. (page 15 lines 479―481).

Round 3

Reviewer 1 Report

All of my comments have been addressed.

This manuscript is a resubmission of an earlier submission. The following is a list of the peer review reports and author responses from that submission.

Round 1

Reviewer 1 Report

Authors are suggested to address the following comments.

  1. Abstract, please highlight the results of proposed work and discuss how it outperforms existing works.
  2. Keywords, more representative keywords should be included to summarize the scope of the manuscript. Also, proper selection of keywords could increase the downloading rate and citation of the article.
  3. Introduction, please clarify if there is relevant related works in this research topic.
  4. Improve the resolution of all figures. They are blurred.
  5. Section 2, Figure 1 should be explained in detail.
  6. Section 2, proper citations are required instead of presenting the URLs in main text.
  7. Section 2, methodology is not clearly explained. Authors have mentioned there are various datasets, each of the datasets may have different settings and nature which may require specific consideration and customized approach.
  8. Figures 2-4, these are abstract and should be explained with the aid of formal network analysis. Parameters and statistics should be presented.
  9. Figure 2 and Figure 4, legends are suggested.
  10. Throughout the manuscript, authors have cited many references, especially in discussion. Discussion should be organized and presented mainly authors vision. Authors may compare the difference between authors’ work and related works. Also, it would be great to highlight what new findings have been concluded.
  11. What is the limitation of the research work? Also, please share future research direction.
  12. After reading the paper, the contribution is vague and authors are suggested to summarize the contribution in introduction, preferably in point-form.

Reviewer 2 Report

Review- Prediction of pharmacological effects of Agastache rugosa against gastritis using a network pharmacology approach

The authors have set out to prove the efficacy of A. rugosa, a plant used in traditional Korean medicine for treating gastritis. They used a network pharmacological approach, combining both bioinformatic approaches and experiments done on mice. Five components have been selected by ADME screening, and were linked to gastritis related genes. Protein interaction map was generated for the related proteins. OVerall the study offers a good design for further study into individual compounds, which could one day yield a potential drug candidate. Several parts of the article are very poorly written and I suggest these mistakes be fixed.

1-27 »inflammatory response in the gastric mucosa by irritation« should be »inflammation response to irritation in the gastric…«

1-29 Repeating that gastritis is inflammation. It would be sufficient to state it once, either in the first or the second sentence.

1-42 Taste and »channel affiliations« are very unrelated and would be better off used separately.

2-64 components were searched »for«. For is missing.

2-78 Previously is in the wrong part of the sentence, it should be »isolated previously from A. rugosa in our laboratory.

3-99 »were searched for« for is missing when referring to searching genes.

3-101 This should be changed as proteins are chemical compounds as well, perhaps between small molecules and proteins and for protein-protein interactions.

3-104 instead of »which indicated« it should be »indicating«

3-109 Instead »of gastritis related genes« it should be »on gastritis« and missing again is searched FOR

3-111 »that genetic functional information« is false or something is missing. It should be »containing functional genetic information«.

3-116 »Well-designed network-based tools have been predicted between potential genes and target disease«. False, if you mean to state that network based tools can predict a connection between potential genes and target disease rewrite this sentence.